# FlySearch:
# Exploring how vision-language models explore

**Adam Pardyl**[1,2,3]     **Dominik Matuszek**[1,2]     **Mateusz Przebieracz**[2]     **Marek Cygan**[4,5]

**Bartosz Zieliński**[2]                    **Maciej Wolczyk**[1]

## Abstract

The real world is messy and unstructured. Uncovering critical information often requires active, goal-driven exploration. It remains to be seen whether Vision-Language Models (VLMs), which recently emerged as a popular zero-shot tool in many difficult tasks, can operate effectively in such conditions. In this paper, we answer this question by introducing FlySearch, a 3D, outdoor, photorealistic environment for searching and navigating to objects in complex scenes. We define three sets of scenarios with varying difficulty and observe that state-of-the-art VLMs cannot reliably solve even the simplest exploration tasks, with the gap to human performance increasing as the tasks get harder. We identify a set of central causes, ranging from vision hallucination, through context misunderstanding, to task planning failures, and we show that some of them can be addressed by finetuning. We publicly release the benchmark, scenarios, and the underlying codebase.

## 1 Introduction

Vision-Language Models (VLMs) have rapidly emerged as state-of-the-art performers in tasks ranging from image captioning [32, 69] to robotics [7, 44]. However, real-world decision-making requires curiosity, adaptability, and a goal-oriented mindset. Nevertheless, the ability of VLMs to operate in realistic, open-ended environments remains largely untested. In this paper, we propose a benchmark to understand and enhance the exploratory capabilities of VLMs. We draw inspiration from the field of Object Navigation (ObjectNav), which focuses on creating embodied agents capable of finding a specific object in a simulated environment and navigating to it. The object may not be visible from the agent's initial perspective, meaning the agent must perform a careful search to locate it.

There are several significant differences between our benchmark FlySearch and existing counterparts. First, we measure the exploration capabilities of VLMs themselves rather than analyzing more complex systems built on them. Understanding these capabilities is important because gathering information is a crucial aspect of the emerging agentic systems [45]. Second, while most ObjectNav benchmarks take place indoors, we focus on finding objects in a large outdoor space using an Unmanned Aerial Vehicle (UAV). This allows us to examine how VLMs explore large, diverse, and unstructured areas and how they control the altitude to change the search granularity. Finally, we focus on a zero-shot open-ended search setting, i.e., a tested method should not make any prior assumptions about the testing environment, the categories of objects, or the *search process itself*. For example, methods that use detectors of specific objects are out of the scope of our research.

---

[1]IDEAS NCBR; [2]Jagiellonian University, Faculty of Mathematics and Computer Science; [3]Jagiellonian University, Doctoral School of Exact and Natural Sciences; [4]University of Warsaw; [5]Nomagic.
Correspondence to: Adam Pardyl `adam.pardyl@doctoral.uj.edu.pl`

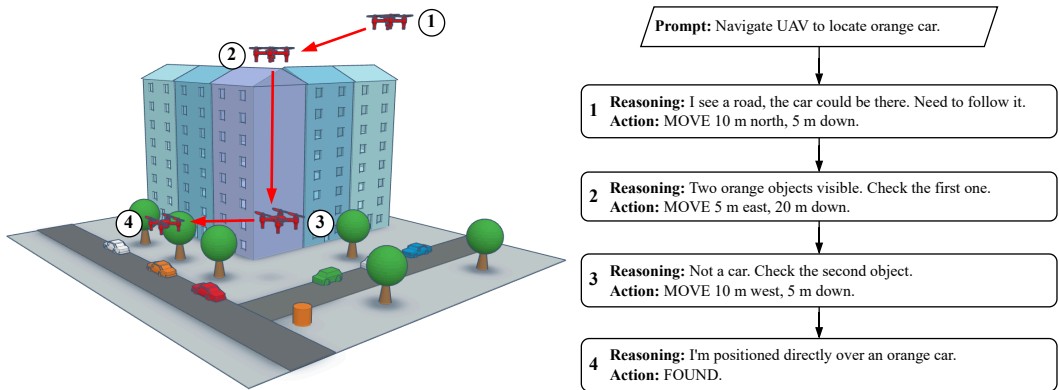

Figure 1: **FlySearch** is a benchmark that evaluates exploration skills using vision-language reasoning. To complete each assessment scenario, a model must locate an object specified in natural language. The agent controls an Unmanned Aerial Vehicle (UAV) by observing images obtained from successive locations of the UAV and providing text commands describing the next move.

FlySearch[1] aims to evaluate the extent to which VLMs can autonomously reason and explore, as well as characterize the limitations of their current capabilities. In FlySearch, the agent (tested method) controls a UAV flying over urban or natural environments to search for objects of interest, such as specific cars, fires, lost people waiting to be rescued, or piles of garbage (see Figure 1). It is built from scratch using the realistic Unreal Engine 5 [15], widely used for video games, providing dynamic 3D scenes with many assets. Moreover, procedural generation allows the generation of an unlimited number of scenarios with varying environmental characteristics, such as time of day, forest density, and UAV launch altitude. The benchmark runs efficiently in headless mode on both modern consumer-grade GPUs (NVIDIA RTX) and standard deep learning clusters (A100/H100), lowering the entry barrier for researchers and practitioners.

FlySearch consists of 3 standardized scenario sets with varying levels of difficulty. **FS-1** tests basic perception and navigation skills, **FS-Anomaly-1** additionally probes the capability of agents to understand the context of the environment, while **FS-2** requires executing a consistent exploration strategy involving a large number of steps to find the object of interest. We evaluate 3 closed-weight and 6 open-weight models on these scenarios and compare the results to scores obtained by humans. We find that VLMs strongly underperform humans on both FS-1 and FS-2. Additionally, while the performance drop between FS-1 and FS-2 is relatively low for humans ($\approx 9\%$), it is massive for VLMs ($\approx 90\%$). Therefore, despite possessing basic navigation and visual comprehension skills, current VLM models fail to form and execute proper exploration strategies, even after GRPO-based fine-tuning.

Our contributions can be summarized as follows:

- We release two high-fidelity outdoor environments built with Unreal Engine 5, enabling realistic and scalable evaluation of embodied agents in complex, unstructured settings.
- We define a suite of object-based exploration challenges designed to isolate and measure the exploration capabilities of VLMs and humans in open-world scenarios.
- We benchmark several popular VLMs in a zero-shot setting and identify consistent failure modes across vision, grounding, and reasoning. Our analysis reveals that these limitations persist even with fine-tuning, suggesting fundamental gaps in current VLM architectures.

## 2 Related work

**ObjectNav.** Our environment is focused on Object-Goal Navigation (ObjectNav or ObjectGoal) task, where the goal is to navigate to a specific object type in a given environment [3, 6]. Using this environment, we instantiate challenges that tie into Language-Driven Zero-Shot ObjectNav [12, 17, 39] tasks, as we expect tested methods to be able to perform search for an arbitrary text-based

---
[1]https://github.com/gmum/FlySearch

Table 1: **Other benchmarks:** We compare FlySearch with other benchmarks focused on VLM evaluation and ObjectNav challenges. We denote cases of partial criterion satisfaction with ✔.

| Environment | Photorealistic | Outdoor | Exploration-focused | 3D | Focus on VLM eval |
|---|---|---|---|---|---|
| Habitat Nav. Challenge [66] | ✔ | ✗ | ✔ | ✔ | ✗ |
| RoboTHOR [11] | ✗ | ✗ | ✔ | ✔ | ✗ |
| AgentBench [35] | ✗ | ✗ | ✗ | ✗ | ✔ |
| LMAct [51] | ✗ | ✗ | ✗ | ✗ | ✔ |
| SmartPlay [63] | ✗ | ✔ | ✗ | ✗ | ✔ |
| BALROG [45] | ✗ | ✔ | ✗ | ✗ | ✔ |
| VisualAgentBench [36] | ✔ | ✔ | ✗ | ✔ | ✔ |
| OpenEQA [40] | ✔ | ✗ | ✗ | ✔ | ✔ |
| FlySearch (ours) | ✔ | ✔ | ✔ | ✔ | ✔ |

object description. Currently, Habitat-Sim [52, 66], AI2-THOR [11, 25], Gibson Env [64] are the most commonly used environments for the ObjectNav task [57]. All of these environments are indoor-focused, while in FlySearch the agent is embodied within a UAV in an outdoor scenario.

**VLN.** There also exists a related broad field of vision-and-language navigation (VLN), mainly concerned with embodied agents following natural language instructions that specify a route from point A to point B [70]. We note the existence of outdoor UAV-centric environments for VLN tasks, such as AerialVLN [34], TRAVEL [61] or CityNav [28]. AerialVLN is an example of a VLN task with high-level UAV control and a simulated environment, which follows the above definition. TRAVEL aims to make drone-based VLN more low-level by modifying the agent's action space for direct UAV control. CityNav introduces a VLN environment based on scans of real-life cities from SensatUrban [21] (instead of simulating them). These papers focus on instruction following in navigation (i.e., VLN), while our benchmark puts emphasis on finding objects without further instructions (i.e., ObjectNav).

**AirSim and OUTDOOR.** Microsoft AirSim [54] is an Unreal Engine 4 plugin that can be used to create UAV simulators, which can also be used for ObjectNav tasks in an outdoor setting. However, to the best of our knowledge, only OUTDOOR [65] considers such application of this software, as ObjectNav is generally dominated by indoor environments. However, even though OUTDOOR considers using UAVs, the action space is still 2-dimensional (horizontal movement only), while FlySearch has a 3-dimensional exploration space (the UAV can also move up and down). This emphasises dynamic altitude control to manage uncertainty and avoid redundant exploration, as the level of visual detail and the size of the visible area varies depending on the altitude. As such, success in FlySearch requires an efficient, emergent search strategy, where the agent must reason about where the object is likely to be, when looking from high altitude. Overall, this is a much more complex problem, requiring pre-existing knowledge of the real world to understand the contextual cues to exploit a wide field of view at high altitudes and fly low only in promising areas.

**AVE.** Additionally, the problem of exploration is considered from a different perspective in Active Visual Exploration (AVE) tasks [46, 47, 53], where a stationary agent is provided with partial observations derived from a large static image, simulating a limited field of view.

**VLM benchmarks.** Vision question-answering (VQA) datasets that contain pairs of images and textual questions are among the most popular approaches to assess the capabilities and limitations of VLMs. There are many examples of VQA-oriented benchmarks, such as [19, 23, 37, 41, 56, 68]. However, to fully evaluate VLM's performance, benchmarks that can measure their performance in practical tasks are needed [22, 33]. Recently, benchmarks that require interaction with an environment have gained popularity in the context of evaluating the capabilites of different VLMs. For example, BALROG [45], VisualAgentBench [36] and LMAct [51] evaluate VLM-based agents on several different environments with visual and textual representations. MineDojo [16] can be used to test different VLM-based agents in an interactive, 3D environment based on the popular *Minecraft* game. Other examples of benchmarks include SmartPlay [63] and AgentBench [35], although these are limited only to the textual modality (with images being "encoded" into natural language). There are also examples of embodied question-answering benchmarks for VLMs, such as OpenEQA [40]. We differ from other benchmarks by specifically focusing on gauging VLMs in the ObjectNav, which allows us to evaluate exploration capabilities of VLMs with a high degree of independence from their other capabilities.

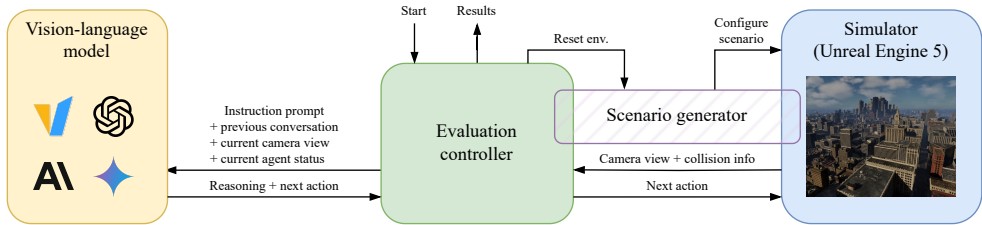

Figure 2: **Evaluation pipeline:** FlySearch consists of three parts besides the vision-language model. The simulator renders near-photorealistic views of a large open-world map and handles basic physics such as collisions. The evaluation controller handles the communication between the evaluated vision-language model and the simulator and performs the evaluation. The scenario generator (functionally integrated with the controller and simulator) procedurally generates new evaluation scenarios.

**VLMs in ObjectNav.** Recently, we have seen a surge in applications of VLM foundation models in ObjectNav [9, 10, 12, 26, 67] and related tasks [38, 62, 72]. However, we note that these papers use VLMs as tools to create more intricate methods and they do not aim to measure the exploration capabilities of VLMs themselves, which are still largely unknown [57]. In this paper, we attempt to gauge VLMs' performance by treating FlySearch as a VLM benchmark meant to compare different models and to understand their shortcomings.

## 3 FlySearch

The goal of FlySearch is to evaluate the visual-spatial reasoning and information-gathering abilities of vision-language models. To achieve this, the model is given control of a simulated multirotor unmanned aerial vehicle equipped with a camera and tasked with finding an object described in natural language, see Figure 1.

### 3.1 Evaluation task

**Environment.** The evaluation environment is a square outdoor area consisting of a fragment of a photorealistic procedurally generated map. The agent starts in the center of the area, at a randomly chosen altitude. Somewhere within the fragment is a target object. The agent must locate it within a limited number of steps and report success.

**Starting prompt.** The model is given a detailed prompt describing its task, including a brief textual description of the object to be located, e.g. *a red pickup truck*. The prompt also describes the communication format, including how to format the responses. We allow the model to preface each of its responses with a description of its reasoning, effectively allowing a chain of reasoning. We provide the prompt template in Appendix G.

**Observation.** At each exploration step, we provide the agent with a $500 \times 500$ pixel RGB image from the simulated UAV camera. To simplify the task, the camera is always facing the ground. The camera image is overlaid with a grid of coordinates to help the model understand movement directions and distances [29]. Additionally, we provide the agent with its height above the ground. In case of FS-2, we also provide the agent with the image specifying how searched object should look like from above; that is done to focus FS-2 more on search. All data except the images is provided in XML format.

**Action.** We assume that the simulated UAV is equipped with an autopilot system. Therefore, the agent does not need to provide any low-level control signals. Instead, it can focus on exploration by providing simple text commands with the tag `<action>(X, Y, Z)</action>`, where each of the coordinates represents a relative position change in meters in the corresponding direction. We introduce a collision avoidance system that stops the movement if an obstacle is detected within a $0.5$ meter radius of the camera position, or if it tries to move out of the fly zone. When the agent decides that it has completed the task, it should respond with the `FOUND` text and end the exploration. If at any point the model response cannot be parsed, the episode is terminated.

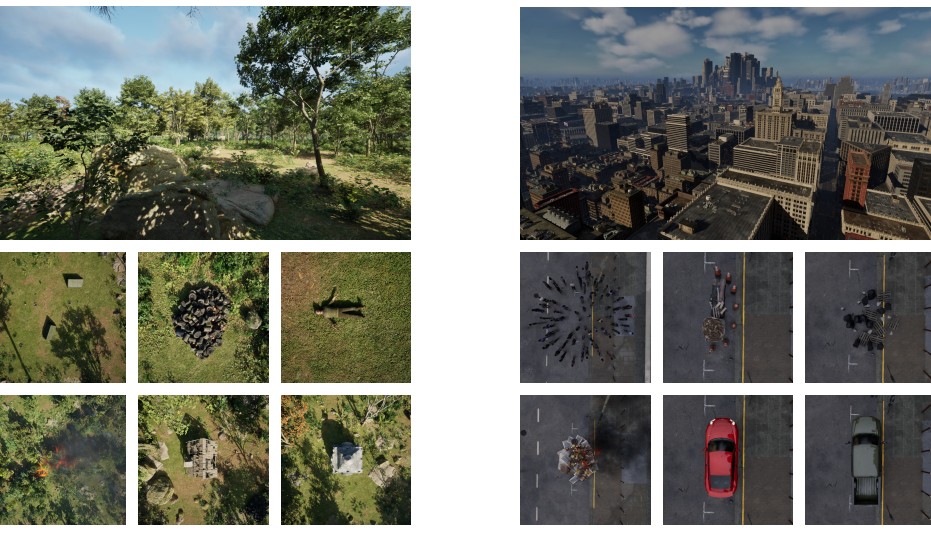

(a) Forest environment.  (b) City environment.

Figure 3: **Environments:** Our benchmark consists of two types of evaluation environments, forest and city. For each environment, we can generate an infinite number of procedurally generated test scenarios. The top row shows a preview of the environment, exhibiting the visual fidelity of the simulation. Below are top-down views of objects, matching the perspective of the agent.

**Metrics.** An episode is considered successfully completed (i.e., the object is found) if the object's center point is within the agent's camera field of view, the agent's position is at most 10 m above the highest point belonging to the object, and the agent returns the FOUND action. This metric is related to the one defined in [6], although we use the altitude difference and field of view instead of the Euclidean distance, as it is easier to estimate for human testers and VLMs – one only needs to make sure that the altitude is not higher than 10 m and the object is visible. We provide details on the implementation of the success criterion in Appendix F.

## 3.2 Evaluation pipeline

The FlySearch system consists of two main components: a simulator based on Unreal Engine 5 and a controller module. The controller is responsible for scenario generation, communication between benchmarked VLMs and the simulator, and result aggregation, as seen in Figure 2.

**Simulator.** To ensure realistic image input for the evaluated tasks, we chose Unreal Engine 5 as the simulation engine. It provides near-photorealistic graphics through real-time ray tracing and dynamic global illumination, while also supporting large, detailed open worlds. The platform is compatible with all major operating systems, and has an open-source codebase, enabling customization for machine learning applications. Additionally, its procedural content generation facilitates environment randomization by integrating parts of the scenario generator directly into the server, allowing us to place tens of thousands of object meshes in seconds. We also leveraged Unreal's extensive online marketplace of free assets to create evaluation scenarios. As such, these assets can be easily used to further extend the benchmark with new environments, scenarios, or objects of interest.

The simulator can be run using any modern consumer-grade graphics card as well as deep learning dedicated solutions (provided Vulkan is supported), and the engine runs in headless/offscreen mode (without a monitor). As such, we found it to perform well on standard computing clusters. Communication between the Unreal Engine simulator and the evaluation controller is handled via standard TCP/IP networking. The simulator-side implementation is provided as a native Unreal Engine plugin. We build upon the UnrealCV project [50], extending its functionality to allow full use of the aforementioned Unreal Engine 5 features.

**Evaluation controller.** The final component of FlySearch is the evaluation controller, implemented in Python. This module oversees the entire lifecycle of the benchmarking process, including setting up scenarios and calculating performance metrics. It also handles communication between the simulator

and the evaluated vision-language model (VLM). Details of the prompts and message templates used in FlySearch are provided in Appendix G. FlySearch supports multiple VLMs, as described in Section 4.1. Additional models can be integrated easily by adding simple adapter code to the controller module or using the open-source vLLM inference server [27].

### 3.3 Evaluation environments

The benchmark consists of two distinct evaluation environment types, a forest and a city, see Figure 3. Each of them has its own set of target objects to find. **Forest environment** is based on the *"Electric Dreams Environment"* Unreal Engine product demo [14]. It consists of sparse forest scenery, including randomly placed rock formations. The map is procedurally generated entirely at runtime by the scenario generator. Moreover, all vegetation on the map is subject to wind changes. **City environment** is a large, modern, American-style city, based on the *"City sample"* Unreal Engine demo [13]. The city layout is a large, semi-procedurally generated map of roughly $4 \times 4$ km. New maps can be generated with the tools provided at build time. Furthermore, random distractor assets (parked cars and walking pedestrians) are spawned by the scenario generator at runtime.

## 4 Evaluation

FlySearch can be used to generate scenarios suited to the user's needs, e.g., finding cars at night in the city or locating stranded people in a very dense forest. However, to facilitate further research and enable fair comparisons, we propose three standardized challenges, i.e., sets of reproducible scenes that can be used to evaluate agents under the same conditions. Although standardized, it's important to highlight that these challenges are not static datasets, as the environment will dynamically react to the actions of the agent.

**FS-1** contains 400 scenes of finding particular objects in the city and forest environments. The agent is explicitly instructed to find an object with a unique description in at most 10 actions. Certain distracting objects may appear, e.g., if we ask the model to find a yellow sports car, there might be cars of other colors or types in the scene. The goal of this challenge is to measure the general search capabilities of the model across a wide array of objects that might be encountered in various UAV applications, e.g., search-and-rescue missions and fire detection. The search area is limited to $400 \times 400 \times 120$ m, and the starting altitude to between 30 and 100 m, and the agent may not fly outside its field of view. We ensure, that the target object is within the field of view of the starting position and its center is not obscured by hard obstacles (it may not be distinguishable due to distance). The agent has to find one of the following:

- in the city: road construction works, crowd, large trash pile, fire, vehicle (variable type).
- in the forest: campsite, trash pile, person, forest fire, building.

In both scenarios, all target objects are positioned on the ground, in semantically correct locations, e.g., a car will be placed in a parking spot, not on the roof. All objects have multiple variations, randomly selected on scenario generation.

**FS-Anomaly-1** is a set of 200 scenes in the city and the forest, where the agent is instructed to find an object that seems out of place, e.g., a giraffe in the city or a UFO in the forest. The goal of this challenge is to measure both the search capabilities of the models as well as their knowledge about what is and is not expected in certain environments. All other settings follow those of the previously mentioned FS-1. Anomalies are placed on the ground level. The anomalous objects to be located are:

- in the forest: UFO (flying saucer), small airplane, helicopter, large dinosaur, airliner,
- in the city: UFO (flying saucer), small airplane, helicopter, medium dinosaur, tank, giraffe.

**FS-2** consists of 200 additional harder scenarios in the city environment. Base setting is the same as in FS-1; however, starting altitude range is raised to between 100 and 125 m and the search are is limited to (-starting altitude, +starting altitude) range in X and Y axis. Moreover, dynamic scene lightning is enabled, simulating different times of day and the maximum allowed altitude is 300 m. Most importantly, we allow the object to be obscured by obstacles (but still visible from the sky) and we allow the model to move beyond its field of view at each step to check models navigation capabilities.

Table 2: **Benchmark results:** Success rates ($\pm$ standard errors) of the evaluated models for FS-1 and FS-2 challenges. We observe that, overall, Gemini 2.0 Flash outperforms all evaluated models. Notably, small open-source models largely fail to solve the test scenarios, while larger models such as the Pixtral 124B achieve better performance. Note that the last row shows the model fine-tuned on the Forest environment (but not the specific FS-1 scenarios), which might lead to overfitting. To signal this, we mark the corresponding result gray.

| Model | FS-1 | | | FS-2 |
| --- | --- | --- | --- | --- |
| | Overall (%) | Forest (%) | City (%) | Overall (%) |
| Human (untrained) | – | – | $66.7 \pm 4.5$ | $60.8 \pm 6.9$ |
| GPT-4o | $39.5 \pm 2.4$ | $45.5 \pm 3.5$ | $33.5 \pm 3.3$ | $3.5 \pm 0.9$ |
| Claude 3.5 Sonnet | $41.2 \pm 2.5$ | $\mathbf{52.0 \pm 3.5}$ | $30.5 \pm 3.3$ | $\mathbf{6.5 \pm 1.2}$ |
| Gemini 2.0 flash | $\mathbf{42.0 \pm 2.5}$ | $42.5 \pm 3.5$ | $\mathbf{41.5 \pm 3.5}$ | $6.0 \pm 1.1$ |
| Phi 3.5 vision | $0.0 \pm 0.0$ | $0.0 \pm 0.0$ | $0.0 \pm 0.0$ | – |
| InternVL-2.5 8B MPO | $2.0 \pm 0.7$ | $2.5 \pm 1.1$ | $1.5 \pm 0.9$ | – |
| Llava-Interleave-7b | $0.8 \pm 0.4$ | $0.0 \pm 0.0$ | $1.5 \pm 0.9$ | – |
| Qwen2.5-VL 7B | $3.8 \pm 1.0$ | $6.0 \pm 1.7$ | $1.5 \pm 0.9$ | $0.0 \pm 0.0$ |
| Qwen2-VL 72B | $17.2 \pm 1.9$ | $16.5 \pm 2.6$ | $18.0 \pm 2.7$ | – |
| Llava-Onevision 72b | $9.5 \pm 1.5$ | $12.5 \pm 2.3$ | $6.5 \pm 1.7$ | – |
| Pixtral-Large | $29.8 \pm 2.3$ | $38.0 \pm 3.4$ | $21.5 \pm 2.9$ | $3.0 \pm 0.8$ |
| Qwen2.5-VL 7B, GRPO on Forest | – | $57.0 \pm 3.5$ | $27.0 \pm 3.1$ | $0.0 \pm 0.0$ |

## 4.1 Baselines

We evaluate a range of popular models. We select three proprietary models: OpenAI GPT-4o (2024-08-06 release) [2, 24], Anthropic Claude 3.5 Sonnet [4], and Google Gemini 2.0 flash (experimental) [18, 58]. Furthermore, we select four small open-weight models, with below 11B parameters: Phi-3.5-vision [1], InternVL2.5-8B-MPO [60], Llava-Interleave-Qwen-7B-dpo-hf [31], and Qwen2.5-VL 7B [5]. Finally, we select three more open-weight models with more than 11B parameters: Qwen2-VL-72B-Instruct [59], Llava-Onevision-Qwen2-72B-ov-hf [30], and Pixtral-Large-Instruct-2411 124B [43]. All models were selected based on their ability to process a full evaluation run, keeping all steps in context. During selection, we tested and rejected Llama-3.2 [42]. Although it architecturally supports handling multiple images, the publicly available model fails to form coherent responses when there is more than one image in the context.

**Human study.** Furthermore, we provide a human baseline for FS-1 City and FS-2 based on a user study of respectively 111 and 51 samples. The study was conducted using an online service, where participants were provided with the benchmark prompt and had to perform the same actions as the VLM. We provide more details about human baselines in Appendix D.

## 4.2 Results

**FS-1.** Table 2 contains the main aggregated results of our study. We find that the state-of-the-art VLMs achieve significantly worse results than non-trained humans on FS-1. While humans score $67\%$ on average, the best-performing Gemini 2.0 manages to find the object in $42\%$ of cases, with Claude and GPT-4o closely following. Large open-weight models fall behind significantly, with Pixtral, the best-performing model in this category, losing 10 percentage points on average to the proprietary models. Finally, the small open-weight models do not show any signal at all, as none of them exceeded 4%. We find that their poor performance can be largely attributed to their inability to follow instructions. Figure 4 shows that small models often do not claim that they have found the object even if it is within range at the end of the episode. As such, we exclude the small models from further analysis. We provide additional results, including measuring the impact of the action representation and the grid overlay, in Appendix H.

**FS-2.** Strikingly, the situation looks much different on FS-2. While on FS-1 humans outperformed best VLMs by $60\%$, on FS-2 this number is closer to $835\%$. We attribute this gap to the lack of systematic exploration abilities in VLMs. Since in FS-1 the object should be visible in the initial

Table 3: **FS-Anomaly-1 results:** Success rates ($\pm$ standard errors) of the evaluated models. Full results are in Appendix H.

| Model | FS-Anomaly-1 Overall (%) |
|---|---|
| GPT-4o | $27.0 \pm 3.1$ |
| Claude 3.5 Sonnet | $27.5 \pm 3.2$ |
| Gemini 2.0 flash | $\mathbf{35.5 \pm 3.4}$ |
| Phi 3.5 vision | $0.0 \pm 0.0$ |
| InternVL-2.5 8B MPO | $3.5 \pm 1.3$ |
| Llava-Interleave-7b | $0.0 \pm 0.0$ |
| Qwen2.5-VL 7B | $2.8 \pm 1.2$ |
| Qwen2-VL-72B | $7.5 \pm 1.9$ |
| Llava-Onevision 72b | $8.5 \pm 2.0$ |
| Pixtral-Large | $15.0 \pm 2.5$ |

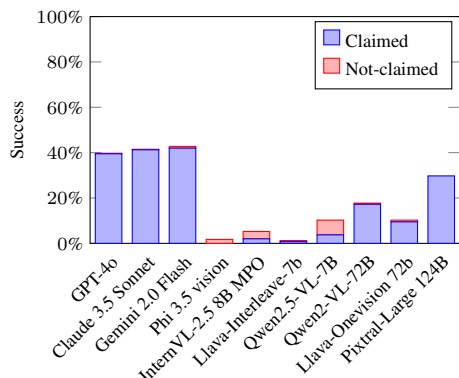

Figure 4: **Not claimed successes in FS-1:** In this figure we compare the number of cases, where the agent located the object, but failed to claim the FOUND action. We observe, that small models often fail to format the text output to report success.

Table 4: Performance of Gemini 2.0 Flash and Pixtral-Large in ablation studies. Surprisingly, increasing the number of steps might lead to worse performance. Additionally, explicitly specifying the object type in FS-Anomaly-1 improves the results as the task becomes easier.

| | Setting | City | | Forest | | Overall | |
|---|---|---|---|---|---|---|---|
| | | Gemini | Pixtral | Gemini | Pixtral | Gemini | Pixtral |
| FS-1 | 5 steps limit | 34.5% | 15.5% | 41.5% | 33.5% | 38.0% | 24.5% |
| | **10 steps limit (baseline)** | **41.5**% | **21.5**% | 42.5% | **38.0**% | **42.0**% | **29.8**% |
| | 20 steps limit | 33.5% | 13.5% | **45.5**% | 36.0% | 39.5% | 24.8% |
| FS-Anomaly-1 | **Searching for an anomaly (baseline)** | 25.0% | 4.0% | 46.0% | 26.0% | 35.5% | 15.0% |
| | Searching for explicit object types | **34.0**% | **7.0**% | **59.0**% | **34.0**% | **46.5**% | **20.5**% |

frame, it requires mostly object recognition and spatial reasoning abilities. On the other hand, since the object might not be within the field of view in FS-2, it requires the agent to implement a strategy that extensively explores the environment over multiple timesteps. While humans intuitively start following the streets in search of the object in question, VLMs mostly wander aimlessly in random directions. The problem is exacerbated due to difficulties with handling long contexts, see the episode length ablation in a later paragraph.

**Fine-tuning results.** We find that although some of these shortcomings can be addressed through simple fine-tuning, the problems with systematic exploration are more fundamental. We finetune Qwen2.5-VL-7B [5] using GRPO [55] on Forest environment in an offline mode, using a synthetic set of randomly generated flight trajectories, see details in Appendix E. The resulting model, presented in the last rows in Table 2, vastly improves upon the base model's performance on FS-1 City scenario, boosting the score by $14$ times, from roughly $1.5\%$ to $21.5\%$. However, the fine-tuning does not impact the results on FS-2 at all, where we still never see any successes.

**Qualitative analysis.** We perform further qualitative analysis of the larger models, see Figure 6 and Appendix J for example trajectories. By manually analyzing failed exploration trajectories on FS-1, we observe that even the most advanced models struggle with spatial reasoning. For instance, when a model loses sight of an object, it often backtracks its moves or starts hallucinating rather than moving toward the object's last known location. In case of FS-2, these issues are aggravated by the additional need of carrying out a systematic search. For example, GPT-4o often flies to the ground and hallucinates the existence of a searched object, whilst not performing any reasonable exploration pattern at all. Moreover, all models fail to handle collisions properly, often attempting to redo the same action when it previously resulted in hitting an object. In one of the trajectories, Gemini states "*I am unable to navigate without hitting the building. I guess I will give up.*"

**Object-specific analysis.** In Figure 5 we zoom in on the specific object classes that appear in FS-1. As expected, classes with large objects, such as buildings have a higher success rate than those with

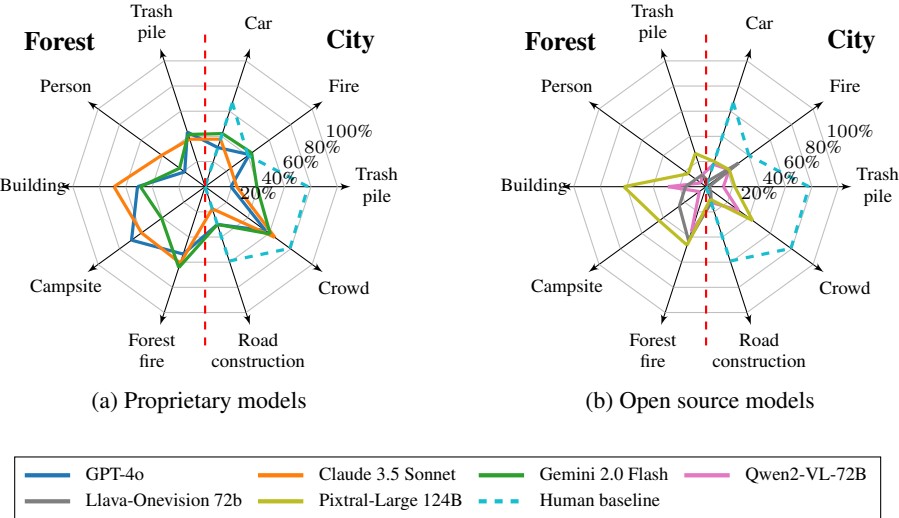

(a) Proprietary models            (b) Open source models

| | | | |
|---|---|---|---|
| —— GPT-4o | —— Claude 3.5 Sonnet | —— Gemini 2.0 Flash | —— Qwen2-VL-72B |
| —— Llava-Onevision 72b | —— Pixtral-Large 124B | - - - Human baseline | |

Figure 5: **Success rate per class:** In this figure, we show the model success rate per each target class for both the Forest and the City environments. Classes with large visible objects such as *Building* for Forest or *Crowd* for City are significantly easier for most models to locate. On the other hand, classes that are difficult to distinguish from the background, such as both garbage classes, are only located by more advanced models. Human baseline for City is marked with dashed line.

Step 1            Step 2            Step 3

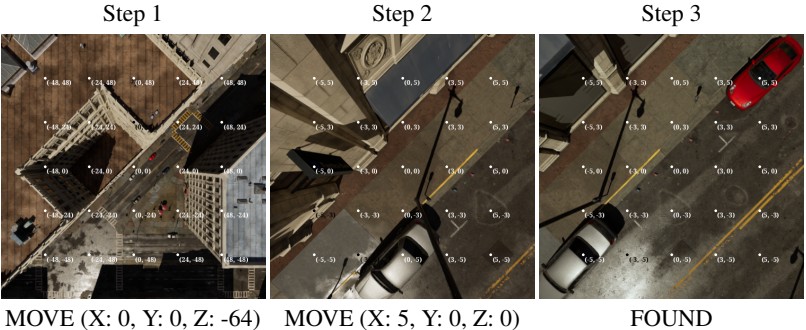

MOVE (X: 0, Y: 0, Z: -64)    MOVE (X: 5, Y: 0, Z: 0)       FOUND

Figure 6: **Example of a successful trajectory in FS-1:** GPT-4o navigates to *red sports car* object by first descending and then moving to the right. The first row shows the model's visual inputs, and the second actions it has taken. Note the presence of the grid overlay on images. Best viewed zoomed in.

smaller figures, such as single persons. However, even finding a building poses a significant challenge since the best-performing Claude 3.5 Sonnet does not manage to find approximately $30\%$ of objects in this class. Interestingly, VLMs tend to be more successful at finding trash piles in the forest rather than in the city, where they are more visible from a distance. Lastly, the road construction site class is one of the hardest classes, even though the name provides a clue as to where the model should look.

**FS-Anomaly-1.** The results for FS-Anomaly-1, presented in Table 3 are on average significantly lower than in FS-1, suggesting that the models struggle with the additional challenge of figuring out which object is out of place. Indeed, as we confirm on a subset of the models, the performance substantially increases once we explicitly name the anomaly object to be found, see Appendix H. Without explicit instructions, more often than not, VLMs find one of many typical, although visually distinct objects to be out of place, while ignoring obvious anomalies. For example, in one of the city scenarios, all closed-source VLMs wrongly identified a *yellow taxi* to be the anomaly, ignoring a *tank* standing next to it. Moreover, we find that the models sometimes tend to misidentify the anomaly objects as something more expected in a given situation, e.g., one of the models assumed that a giraffe walking around city streets is a dog.

**Impact of the number of steps.** Finally, we study the impact of changing the length of each episode in FS-1, i.e., the number of actions that can be taken before the trajectory automatically ends in failure. We find that reducing the number of steps from the baseline 10 to 5 reduces the results by 10% (4pp) for Gemini and 17% (5pp) for Pixtral. More interestingly, we also observe performance deterioration as we increase the step limit to 20 – Gemini's performance falls by 6% (2.5pp) and Pixtral's by 17% (5pp). Breaking these results by category, we find that increasing the number of steps only leads to significant deterioration in the visually cluttered City environment. We discover that models tend to fail when expected to reason and gather information over longer timeframes, which we also observed in FS-2. See more details in Table 4 and Appendix H.

## 5   Conclusion

In this paper, we introduce FlySearch, a dynamic benchmark designed for evaluating exploration capabilities of VLMs. Navigating three-dimensional environments and finding objects of interest are everyday real-world tasks that remain underrepresented in VLM benchmarking. To address this gap, FlySearch leverages Unreal Engine 5, a highly realistic video game engine to procedurally generate scenarios of searching for objects in urban and natural environments. Using the three standardized challenges, FS-1, FS-Anomaly-1, and FS-2, we show that VLMs underperform compared to human baseline, especially when it comes to more complex exploration tasks. At the same time, this study has certain limitations that offer interesting directions for future work. In this paper, we purposefully avoid testing more sophisticated ObjectNav methods [9, 26, 67], since our main focus lies in understanding pure VLM capabilities. At the same time, checking their performance in FlySearch could bring interesting insights to the field. Additionally, we use a simple prompting technique, and one could possibly get better results out of VLMs by leveraging few-shot learning [8, 48] or prompt optimization tools [49, 73].

## Acknowledgments

This paper has been supported by the Horizon Europe Programme (HORIZONCL4-2022-HUMAN-02) under the project "ELIAS: European Lighthouse of AI for Sustainability", GA no. 101120237. This research was funded by National Science Centre, Poland (grant no. 2023/50/E/ST6/00469 and Sonata Bis grant no 2024/54/E/ST6/00388). The research was supported by a grant from the Faculty of Mathematics and Computer Science under the Strategic Programme Excellence Initiative at Jagiellonian University. We gratefully acknowledge Polish high-performance computing infrastructure PLGrid (HPC Center: ACK Cyfronet AGH) for providing computer facilities and support within computational grant no. PLG/2024/017483. Some experiments were performed on servers purchased with funds from the Priority Research Area (Artificial Intelligence Computing Center Core Facility) under the Strategic Programme Excellence Initiative at Jagiellonian University.

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

# A Impact Statement

In this paper, we focus on the abstract problem of visual exploration – how to interact with the environment to locate the objects of interest. To evaluate this ability in VLMs, we propose a benchmark designed around flying a UAV to find objects in urban and natural environments. This is relevant to real-world applications with positive societal impact, such as search-and-rescue missions, forest fire detection, or personal assistance. However, autonomous UAVs also pose risks as they can be misused by bad agents for surveillance or military operations. We implore users to use their best judgment for the use of the benchmark. Our benchmark is not designed for surveillance or military applications. We urge other researchers to ensure that their benchmarks (including FlySearch derivatives) and models are not directly or indirectly optimized for malicious objectives. We encourage researchers to evaluate their systems for potential biases or unintended behaviors that could lead to misuse. Finally, we would like to highlight the importance of regulatory oversight and the need for clear ethical guidelines in the deployment of autonomous UAVs.

# B Funding Transparency Statement

**Funding.** All financial activities supporting the submitted work are listed in the Acknowledgments section of the main paper.

**Competing Interests.** At the time of the publication, Maciej Wołczyk is working at Google. However, he did the majority of the work on this paper while at IDEAS NCBR and has not been involved in benchmarking Google models. The authors declare that they have no competing interests and no financial relationships with any entity that could be perceived as influencing the content of this work.

# C Benchmark specification

In this section we present the parameters of benchmark scenario generation and evaluation details.

## C.1 Scenario parameters

Table S1: **Parameters in scenario generation:** In this table, we present value ranges for variables that have an impact on the search trajectory. We sample from them uniformly at random while creating a scenario. FS-Anomaly-1 has identical value ranges to FS-1, except for the searched object type and searched object asset. Furthermore, in FS-1 and FS-Anomaly-1 there will be a clear line of sight from the starting position of the agent to the object. In FS-2, we relax this condition and only validate that the object is not under an obstacle (e.g. is not hidden under a bridge, but can be behind a building). Lastly, objects can be placed anywhere in the forest environment and in one of 31852 semantically correct locations in the city environment.

| Parameter | FS-1 | | FS-2 |
| --- | --- | --- | --- |
| | Value range (Forest) | Value range (City) | Value range (City) |
| Seed | $\mathbb{N} \cap [0, 1000000000]$ | $\mathbb{N} \cap [0, 1000000000]$ | $\mathbb{N} \cap [0, 1000000000]$ |
| Agent's starting height ($h$) [m] | $\mathbb{N} \cap [30, 100]$ | $\mathbb{N} \cap [30, 100]$ | $\mathbb{N} \cap [100, 125]$ |
| Agent's starting position offset [m] | $[-0.5h, 0.5h]$ | $[-0.5h, 0.5h]$ | $[-0.95h, 0.95h]$ |
| Searched object type | {campsite, trash, person, fire, building} | {construction works, crowd, trash, fire, car} | |
| Searched object coordinates | Anywhere on the map | 38152 possible placements | |
| Searched object asset | Dependent on the object type | Dependent on the object type | |
| Sun elevation angle | $[10, 90]°$ over horizon | $45°$ over horizon | $[10, 90]°$ over horizon |
| Sun azimuth angle | $[0, 360]°$ | $110°$ | $[0, 360]°$ |
| Tree density | $[0.0, 0.3]$ | N/A | N/A |
| Rock density | $[0.0, 0.1]$ | N/A | N/A |
| Object visibility | Visible from agent's starting position | | Not under an obstacle |

We present all variables that can be used to generate a new scenario in Table S1, such as whether the searched object should be visible from the UAV's initial location. In Table S2, we describe additional parameters that do not impact the scenario generation, but govern details of evaluation – such as:

- Search area bounds – a rectangle centered on agent's initial position, describing bounds inside of which agent can move. Note that this only serves as a limitation on agent's movement and does not impact the scenario generation process,

- Maximum altitude – if an agent's action would bring it above that threshold, it is considered invalid and the agent is asked to issue a new instruction,

- Whether agent can move beyond its *current* view – in FS-1 and FS-Anomaly-1 the agent is prevented from leaving visible area, discouraging hallucination,

- Number of available actions,

- Number of possible *consecutive* retries if agent's action is considered invalid and needs to be redone,

- Whether the agent should also receive an image showcasing how the target object should look like from above.

Table S2: **Additional parameters:** In this table, we present configurations used during evaluations in FS-1, FS-Anomaly-1 and FS-2. Number of possible retries in case of invalid action was introduced to prevent LLMs from "hanging" by constantly performing invalid actions. As such, this limit was not present while evaluating human baselines. Similarly, we allow humans to fly out of their current view even in FS-1. We note that FS-1 and FS-Anomaly-1 use the exact same configuration, while more search-oriented FS-2 uses a slightly different one. In this table, $h$ denotes starting height of the UAV.

| Parameter | FS-1 / FS-Anomaly-1 | FS-2 |
|---|---|---|
| Search area bounds [m] | $400 \times 400$ | $2h \times 2h$ |
| Maximum altitude [m] | 120 | 300 |
| Agent can move beyond its current view | No (Yes for humans) | Yes |
| Number of available actions | 10 | 20 |
| Number of possible retries if action is invalid | 5 ($\infty$ for humans) | 5 ($\infty$ for humans) |
| Object type specification modality | Text | Text + Image |

## C.2 Target details

The targets in the City environment are as follows:

- Road construction works – a construction zone at the side of a road,

- Crowd – a group of over 30 randomly generated people standing close together,

- Large trash pile – a random pile of trash (bags, car tires, barrels, metal sheets, etc.),

- Fire – a burning car or pile of trash with fire and smoke,

- Car – a randomized car or truck with a specific color (we provide the agent with the color and type).

While in the Forest the goal is to find:

- Campsite – a randomly generated set of camping equipment, including at least one tent,

- Trash pile – a pile of car tires, barrels, metal scrap, and other waste (different than in the forest environment),

- Person – a human laying flat on the ground, representing an injured hiker,

- Forest fire – a burning forest area, emitting a large cloud of smoke,

- Building – an abandoned building in various styles.

## D   Human baselines

In order to evaluate human performance on FlySearch we provide a web-based interface for human testers. The interface is designed to be consistent with VLM evaluation procedure, while allowing for efficient human interaction. Therefore, instead of using XML formatted text for communication, the interface uses standard HTML forms and buttons. In Fig. S1 the welcome screen is presented.

The user is shown the model prompt (with XML formatting instructions omitted) and can select the between FS-1 and FS-2. In Fig. S2 and Fig. S3 the communication screens for FS-1 and FS-2 respectively are shown. On each step the evaluator has to either fill in X, Y, Z action coordinates and press the MOVE button, or if they believe they have fulfilled the task criterion they have to press the FOUND button. Finally, after clicking FOUND or exhausting possible moves the user is notified if they have succeeded or failed the scenario. During the entire evaluation, users can report any issues with a built-in bug report form.

Source code of the interface and associated back-end server is included in FlySearch codebase. The study was conducted with participation of human manual software testers (employees). The study did not pose any risks to participants. No Personally Identifiable Information was gathered in the process. The study was conducted according to institutional regulations on data gathering and processing and was subject to internal approval process.

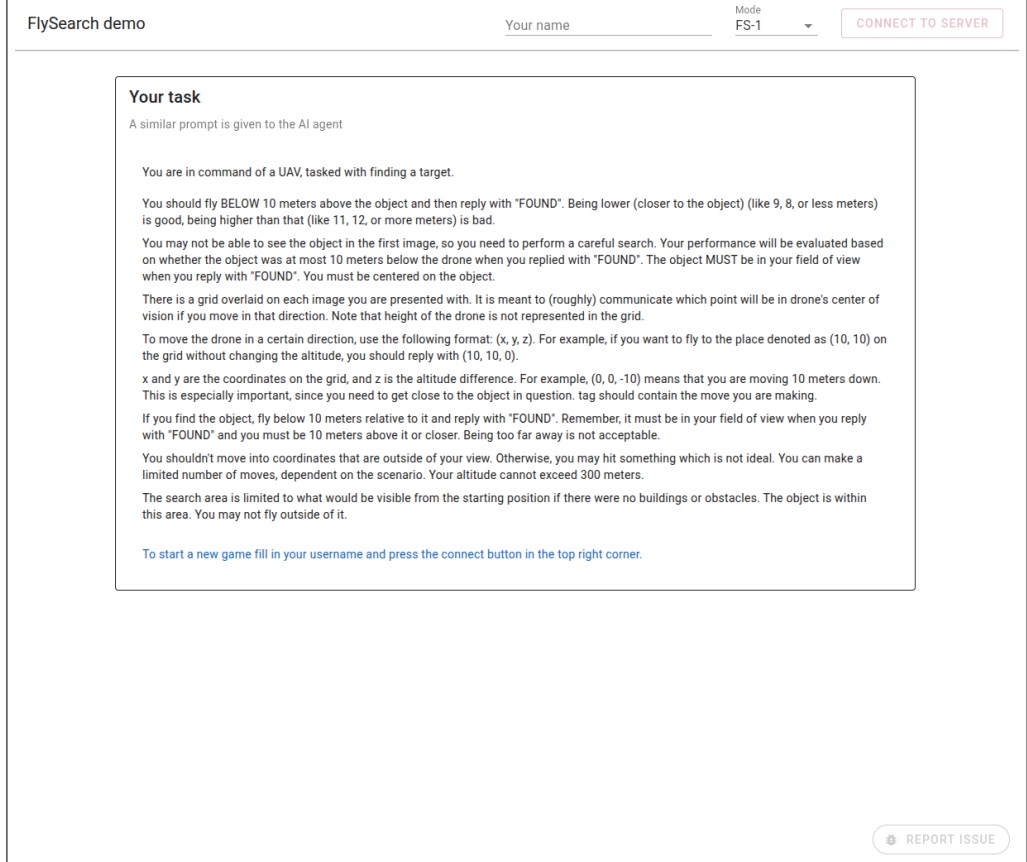

Figure S1: **Human study – start screen**: Screenshot of the FlySearch human interface welcome screen, containing instructions derived from the benchmark prompt. The screen also contains a field for the participant identifier (nickname), a scenario generator switch (FS-1/FS-2), connection button and bug report button.

# E   Fine-tuning details

To provide a reasonable baseline score for fine-tuning VLMs for spatial reasoning we train the Qwen VL 2.5 7B [5] model on the Forest environment and evaluate it on the City environment in both FS-1 and FS-2. The Qwen VL 2.5 7B model is selected as it is the best performing model in the *small open-source model* category in our evaluation. Since the City and Forest environments share almost no graphical assets and are vastly different visually, training the model on Forest allows for a fair comparison with other models on the City environment. That is, the model cannot learn to visually recognize relevant objects and their placement in the evaluated scenario.

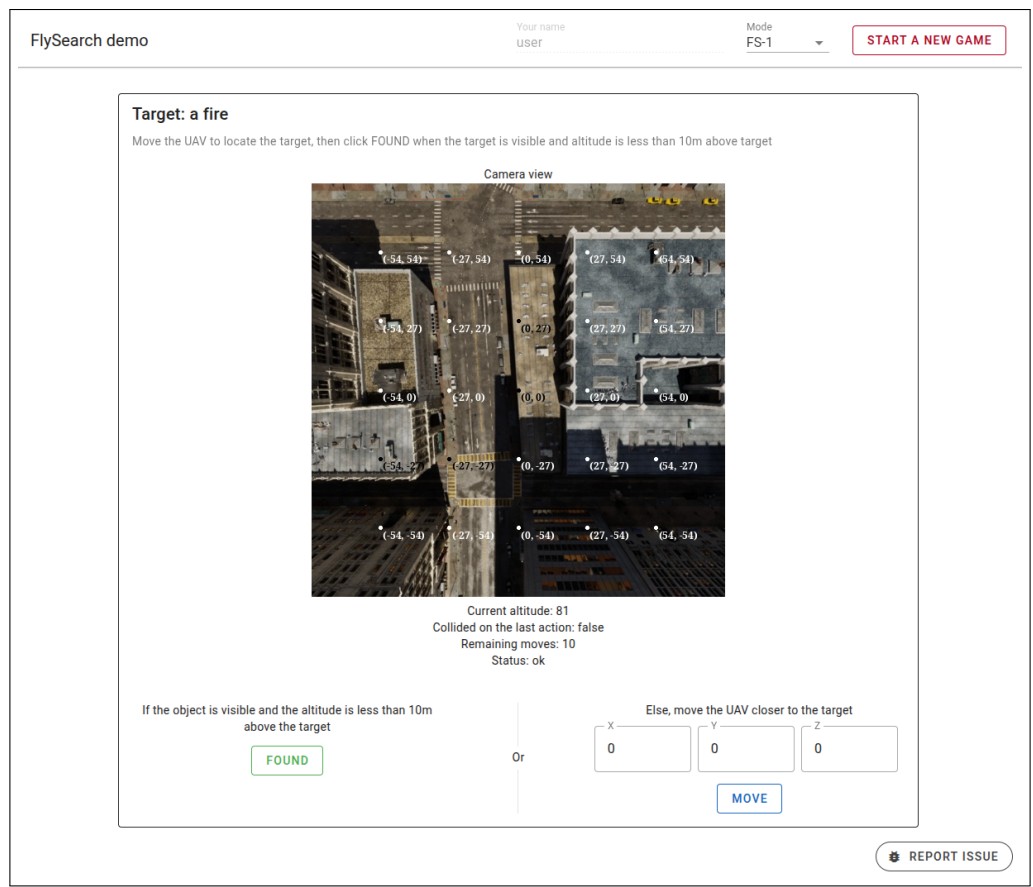

Figure S2: **Human study – FS-1 screen**: Screenshot of the FlySearch human interface FS-1 screen. The page contains the target description, agent camera view, altitude, object and boundary collision information, and action controls. The image is overlaid with grid coordinates, exactly as the image provided to the evaluated VLM. All state and action elements are the same as those provided to the model, but for the formatting (a form instead of XML formatted text).

Using the Forest environment, we generate 6750 unique exploration scenarios. In each scenario, we pre-record a flight trajectory from the starting point to the target. The trajectory is created by sampling 30 evenly spaced steps over a straight line. To each flight step a random position offset in range of $[-10, 10]$ m, and capture camera views using our simulator. Finally, we augment the dataset by generating 10 new trajectories from each captured episode, randomly dropping some of the flight steps. Resulting trajectories have between 1 and 10 steps, depending on the distance between start and target. Each trajectory is further cut at random step and formatted as a conversation to be completed by the VLM, resulting in 67500 training samples.

Standard supervised fine-tuning does not provide sufficient improvement in results (11.1% on FS-1 City, compared to 1.5% of the base model). This is likely due to the fact, that each scenario can be solved in multiple ways, not just the ideal trajectory. Therefore, we apply GRPO fine-tuning [55] in step-wise, offline manner. That is, using the pre-generated training dataset we train the model to predict one next action. We define the reward function for the step with the following pseudo-code:

Listing 1: Reward for GRPO fine-tuning (pseudo-code).

```
def reward(model_output):
    if model_output is not parsable:
        return 0

    reasoning, action = parse(model_output)

    reasoning_reward = min(1, len(reasoning) / 100)
```

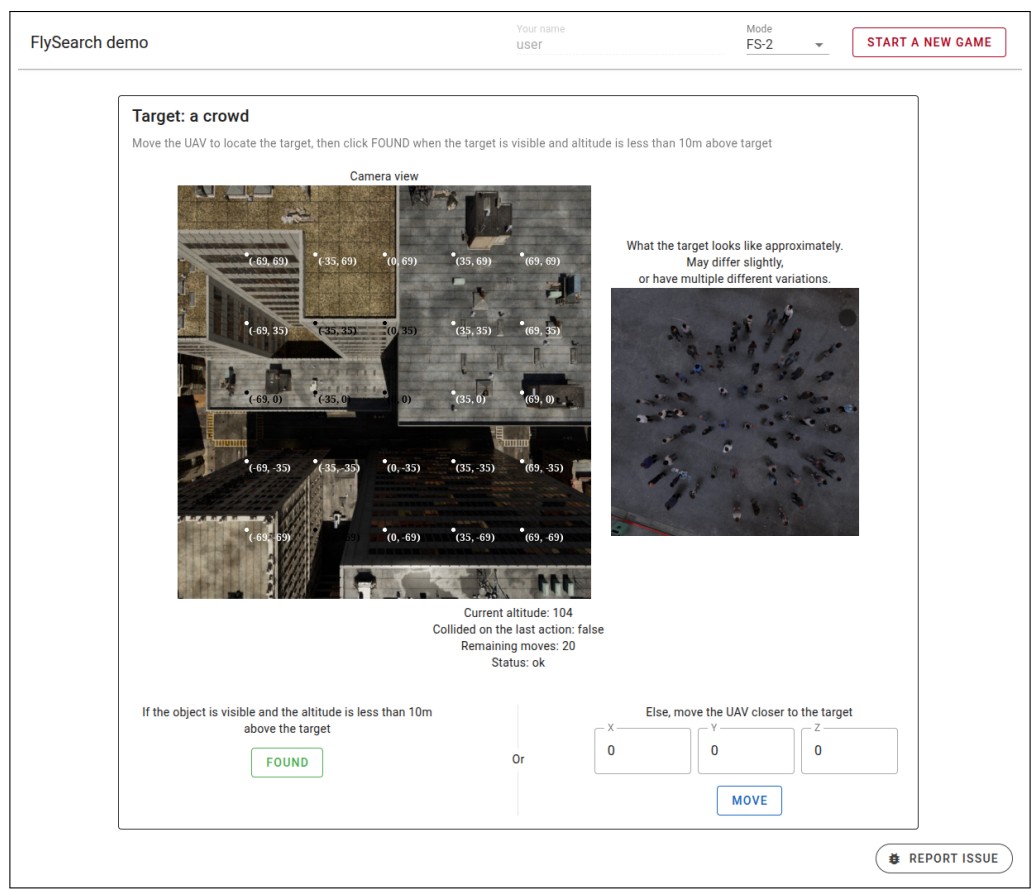

Figure S3: **Human study – FS-2 screen**: Screenshot of the FlySearch human interface FS-2 screen. The page contains the same elements as for FS-1 plus an additional visual description of the target (a neutral background and lightning picture, not an actual image of the searched object).

```
if action == 'FOUND':
    if target_located:
        return 1 + reasoning_reward
    else:
        return 0 + reasoning_reward

action_reward = (current_distance - next_distance) / current_distance
action_reward = max(min(action_reward, 1), 0) # clip between 0 and 1

if target_located:
    # action should be FOUND, decrease reward for further moves
    mod = max(0., (altitude - targed_height compare)) / 10
    action_reward = (mod ** .5) * action_reward

return reasoning_reward + action_reward
```

Therefore, the model is incentivized to both provide at least 100 tokens of reasoning output and to move closer to the target in each step. When the model can respond with FOUND action the reward for further moves towards the target is decreased. Finally, if the agent responds with FOUND it receives a binary reward based on the success.

We perform the GRPO fine-tuning using the Swift library [71], and use LoRA [20] to speed up the training. Moreover, we freeze the vision encoder part of the model to prevent overfitting on visual features. Bellow are the full parameters of the training:

Listing 2: GRPO fine-tuning command.

```
NPROC_PER_NODE=2 CUDA_VISIBLE_DEVICES=0,1,2,3 swift rlhf \
  --rlhf_type grpo \
  --model Qwen/Qwen2.5-VL-7B-Instruct \
  --train_type lora \
  --dataset $DATA_PATH \
  --num_train_epochs 1 \
  --per_device_train_batch_size 8 \
  --per_device_eval_batch_size 8 \
  --learning_rate 1e-5 \
  --gradient_accumulation_steps 1 \
  --eval_steps 100 \
  --save_steps 100 \
  --save_total_limit 2 \
  --logging_steps 5 \
  --dataloader_num_workers 4 \
  --attn_impl flash_attn \
  --use-hf \
  --torch_dtype bfloat16 \
  --deepspeed zero2 \
  --target_modules all-linear lm_head \
  --lora_alpha 32 \
  --lora_rank 8 \
  --external_plugins rewardplugin.py \
  --reward_funcs fly_search \
  --max_completion_length 1024 \
  --warmup_ratio 0.05 \
  --num_generations 16 \
  --dataset_num_proc 4 \
  --temperature 0.9 \
  --log_completions true \
  --use_vllm true \
  --vllm_gpu_memory_utilization 0.9 \
  --vllm_limit_mm_per_prompt '{"image": 10}' \
  --num_infer_workers 2 \
  --async_generate true \
  --seed 42 \
  --split_dataset_ratio 0
```

The entire process takes several hours using 4 NVIDIA H100 GPUs. After fine-tuning the model is evaluated with standard FS-1 City and FS-2 settings, achieving 27.0% and 0.0% accordingly. Moreover, it achieves 57.0% accuracy on the Forest environment it was trained upon.

## F  Success criterion implementation

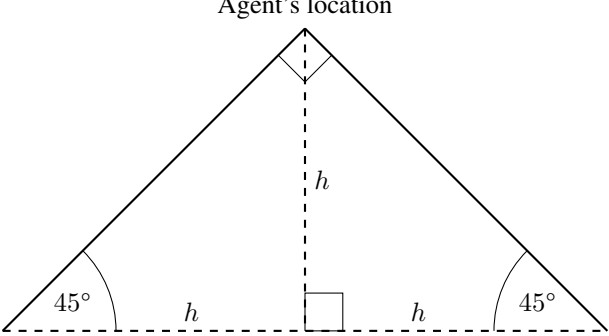

Figure S4: **Success criterion illustration:** The searched object's center must be inside of the camera's cone of view. We assume the camera's field of view is 90°.

For a trajectory to be considered successful, at the end of it the agent must be located less than 10 meters above the object's highest point. Furthermore, the object must be seen from the agent's location.

To check whether the agent can see the object, we calculate agent's cone of view and check whether object's center is inside of the cone of view. This calculation is straightforward, because camera's field of view is $90°$. It easily follows that agent at altitude $h$ sees an area of $2h \times 2h$ meters. We provide a drawing to illustrate that fact in Fig. S4.

In some cases, this implementation may cause misclassifications as a failure, due to the agent „seeing" an object in a human sense (e.g. a significant part of it's edges), but not having object's center in its cone of view. To avoid this issue, we specify in the prompt that agent should be *centered* on the object.

# G  Prompt templates

In this section we present the prompt templates for all benchmark tasks.

## G.1  Prompt template for FS-1 and FS-Anomaly-1

Listing 3: Prompt template for FS-1 and FS-Anomaly-1.

```
<Context>
    You are in command of a UAV, tasked with finding {TARGET}.
</Context>

<Objective>
    You should fly BELOW 10 meters above the object and then reply with "FOUND".
     Being lower (closer to the object) (like 9, 8, or less meters) is good, being
     higher than that (like 11, 12, or more meters) is bad.

    You may not be able to see the object in the first image, so you need to perform
      a careful search. Your performance will be evaluated based on whether the
     object was at most 10 meters below the drone when you replied with "FOUND". The
      object MUST be in your field of view when you reply with "FOUND". You must be
     centered on the object.
</Objective>

<Coordinates>
    There is a grid overlaid on each image you are presented with. It is meant to (
     roughly) communicate which point will be in drone's center of vision if you
     move in that direction. Note that height of the drone is not represented in the
      grid.
</Coordinates>

<Controls>
    <Action space>
        To move the drone in a certain direction, use the following format:
        <Action>(x, y, z)</Action>. For example, if you want to fly to the place
     denoted as (10, 10) on the grid without changing the altitude, you should reply
      with <Action>(10, 10, 0)</Action>.

        x and y are the coordinates on the grid, and z is the altitude difference.
     For example, <Action>(0, 0, -10)</Action> means that you are moving 10 meters
     down. This is especially important, since you need to get close to the object
     in question.

    </Action space>

    <Formatting>

        Your each response should contain XML <Reasoning> tag and <Action> tag.
        <Reasoning> tag should contain your reasoning for the move you are making.
```

```
    <Action> tag should contain the move you are making.

    If you find the object, fly below 10 meters relative to it and reply with "
    FOUND". Remember, it must be in your field of view when you reply with "FOUND"
    and you must be 10 meters above it or closer. Being too far away is not
    acceptable.

    For example:

    <Reasoning>This yellow point might be the object in question. I need to go
    lower to check for that. If it's not the object in question, I will continue
    the search. I will also slightly go to the north.</Reasoning>
    <Action>(5, 0, -30)</Action>

  </Formatting>

  <Limitations>
    You shouldn't move into coordinates that are outside of your view. Otherwise,
     you may hit something which is not ideal.
    You can make at most 9 moves. Your altitude cannot exceed 120 meters. Your
   search area is 400x400m from the drone's starting position.
    </Limitations>
</Controls>
```

## G.2   Prompt template for FS-2

Listing 4: Prompt template for FS-2.

```
<Context>
   You are in command of a UAV, tasked with finding {TARGET}.
</Context>

<Objective>
   You should fly BELOW 10 meters above the object and then reply with "FOUND".
    Being lower (closer to the object) (like 9, 8, or less meters) is good, being
    higher than that (like 11, 12, or more meters) is bad.

   You may not be able to see the object in the first image, so you need to perform
    a careful search. Your performance will be evaluated based on whether the
    object was at most 10 meters below the drone when you replied with "FOUND". The
     object MUST be in your field of view when you reply with "FOUND". You must be
    centered on the object.
</Objective>

<Coordinates>
   There is a grid overlaid on each image you are presented with. It is meant to (
    roughly) communicate which point will be in drone's center of vision if you
    move in that direction. Note that height of the drone is not represented in the
     grid.
</Coordinates>

<Controls>
   <Action space>
      To move the drone in a certain direction, use the following format: <Action
    >(x, y, z)</Action>. For example, if you want to fly to the place denoted as
    (10, 10) on the grid without changing the altitude, you should reply with <
    Action>(10, 10, 0)</Action>.

      x and y are the coordinates on the grid, and z is the altitude difference.
    For example, <Action>(0, 0, -10)</Action> means that you are moving 10 meters
    down. This is especially important, since you need to get close to the object
    in question.

   </Action space>
```

```
<Formatting>

    Your each response should contain XML <Reasoning> tag and <Action> tag.
    <Reasoning> tag should contain your reasoning for the move you are making.
    <Action> tag should contain the move you are making.

    If you find the object, fly below 10 meters relative to it and reply with "
FOUND". Remember, it must be in your field of view when you reply with "FOUND"
and you must be 10 meters above it or closer. Being too far away is not
acceptable.

    For example:

    <Reasoning>This yellow point might be the object in question. I need to go
lower to check for that. If it's not the object in question, I will continue
the search. I will also slightly go to the north.</Reasoning>
    <Action>(5, 0, -30)</Action>

</Formatting>

<Limitations>
    You shouldn't move into coordinates that are outside of your view. Otherwise,
 you may hit something which is not ideal.
    You can make at most {glimpses - 1} moves. Your altitude cannot exceed 300
meters.

    The search area is limited to what would be visible from the starting
position if there were no buildings or obstacles. The object is within this
area. You may not fly outside of it.
    </Limitations>
</Controls>
```

Furthermore, along with the textual description of the searched object we pass the image, showcasing how it should look like from above, with a following annotation:

Listing 5: Annotation of searched object's image in FS-2.

```
The object you're looking for is similar to this. This is NOT the drone's current
    view.
```

# H    Additional Experiments

In this section we present additional experimental results, evaluating design choices of our benchmark.

Table S3: Performance of Gemini 2.0 Flash and Pixtral-Large on ablations.

| | Setting | City | | Forest | |
|---|---|---|---|---|---|
| | | **Gemini** | **Pixtral** | **Gemini** | **Pixtral** |
| FS-1 | **Baseline** | **41.5**% | **21.5**% | **42.5**% | **38.0**% |
| | Compass actions | 17.5% | 21.0% | 17.5% | 22.0% |
| | No grid overlay | 17.0% | 15.5% | 31.5% | 20.0% |
| FS-Anomaly | **Baseline** | 25.0% | 4.0% | 46.0% | 26.0% |
| | Anomaly with ID | **34.0**% | **7.0**% | **59.0**% | **34.0**% |

**Impact of the action space.** We test how changing the action format affects performance by replacing the default Cartesian $(x, y, z)$ movements with compass-style commands that specify a direction (north, south, east, west, up, down) and distance. This type of action space is common in previous works. As shown in Table S3, this change significantly degrades performance for both

Table S4: **FS-Anomaly-1 results:** Success rates ($\pm$ standard errors) of the evaluated models.

| Model | FS-Anomaly-1 | | |
|---|---|---|---|
| | Overall (%) | Forest (%) | City (%) |
| GPT-4o | $27.0 \pm 3.1$ | $39.0 \pm 4.9$ | $15.0 \pm 3.6$ |
| Claude 3.5 Sonnet | $27.5 \pm 3.2$ | $37.0 \pm 4.9$ | $18.0 \pm 3.9$ |
| Gemini 2.0 flash | $\mathbf{35.5 \pm 3.4}$ | $\mathbf{46.0 \pm 5.0}$ | $\mathbf{25.0 \pm 4.4}$ |
| Phi 3.5 vision | $0.0 \pm 0.0$ | $0.0 \pm 0.0$ | $0.0 \pm 0.0$ |
| InternVL-2.5 8B MPO | $3.5 \pm 1.3$ | $6.0 \pm 2.4$ | $1.0 \pm 1.0$ |
| Llava-Interleave-7b | $0.0 \pm 0.0$ | $0.0 \pm 0.0$ | $0.0 \pm 0.0$ |
| Qwen2.5-VL 7B | $2.8 \pm 1.2$ | $3.7 \pm 2.1$ | $2.0 \pm 1.4$ |
| Qwen2-VL-72B | $7.5 \pm 1.9$ | $10.0 \pm 3.0$ | $5.0 \pm 2.2$ |
| Llava-Onevision 72b | $8.5 \pm 2.0$ | $11.0 \pm 3.1$ | $6.0 \pm 2.4$ |
| Pixtral-Large | $15.0 \pm 2.5$ | $26.0 \pm 4.4$ | $4.0 \pm 2.0$ |

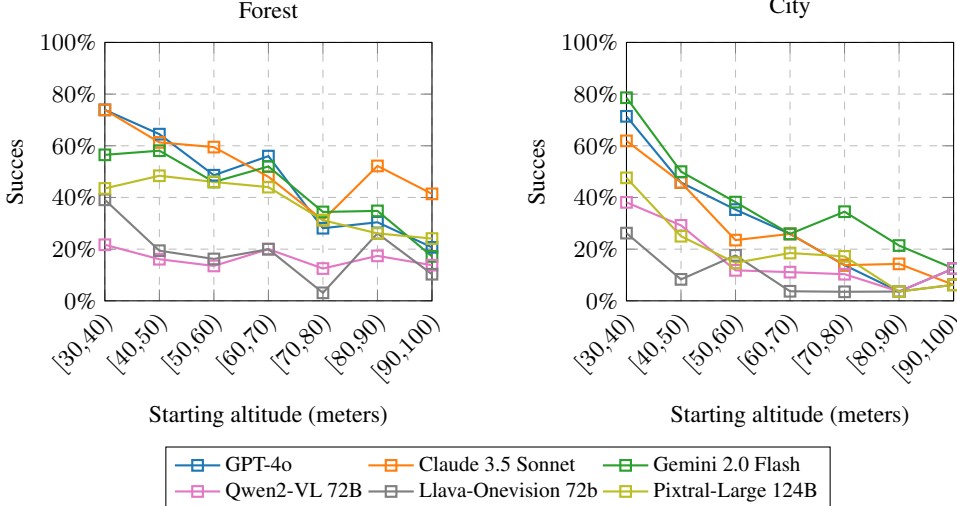

Figure S5: **Success rate per starting altitude interval:** In this figure, we plot the success rates of the models as a function of the initial altitude interval for both the Forest and the City environments in the FS-1 challenge. The performance drop that comes with increasing altitude is more pronounced for the City environment.

Gemini 2.0 Flash and Pixtral-Large. Gemini's accuracy drops from 42.5% to 17.5% in Forest and from 41.5% to 17.5% in City. Pixtral falls from 38% to 22% in Forest, with little change in City (21.5% to 21%). These results highlight the importance of a flexible, continuous action space. Compass-style movement restricts fine-grained control and appears to limit the models' ability to effectively search.

**Impact of the grid overlay.** To assess the importance of the grid overlay on each glimpse, we run an ablation where glimpses are shown without the grid. As shown in Table S3, removing the grid leads to a notable drop in performance for both Gemini 2.0 Flash and Pixtral-Large. Gemini's success rate falls from 42.5% to 31.5% in Forest, and more dramatically in City—from 41.5% to 17%. Pixtral shows a similar trend, dropping from 38% to 20% in Forest and from 21.5% to 15.5% in City. These results suggest that the grid plays a key role in helping models maintain spatial awareness, especially in more complex environments. Without it, both localization and goal-oriented planning appear to suffer. This result is consistent with previous works on vision-language coordination [29].

**Impact of hidden vs explicit anomaly categories.** In the FS-Anomaly setting, the model must identify an object that doesn't belong in the surrounding environment. In the base version, the

anomaly category is implicit, requiring the model to infer what is out of place. In this ablation, we explicitly tell the model what object to look for.

As shown in Table S3, providing the anomaly identity significantly improves performance. Gemini's accuracy increases from 46% to 59% in Forest and from 25% to 34% in City. Pixtral also improves, from 26% to 34% in Forest and from 4% to 7% in City.

This result suggests that models often struggle with anomaly detection due to semantic ambiguity rather than visual perception. Explicitly stating the anomaly class helps disambiguate the task and leads to more focused and successful exploration.

**FS-Anomaly-1 full results.** We provide full results of the FS-Anomaly-1 benchmark with distinction between Forest and City environment performance in Table S4.

**Impact of the starting height.** Finally, we verify how the results vary depending on the starting height. As seen in Figure S5, the success rate decreases as the starting altitude (and therefore distance to the target) increases. This is particularly significant in the City environment, where the success rate for the best models drops by half, likely due to the high number of distracting objects. In contrast, in the generally easier Forest environment, this decline is less pronounced and occurs mainly in the upper half of the altitude range.

## I  Sim2real gap

The main design goal of FlySearch is the evaluation of VLM exploration and 3D spatial reasoning capabilities, with UAV object navigation serving as a realistic assessment task. However, we acknowledge the fact that FlySearch can serve as a platform for testing methods designed specifically for UAV control, including non-VLM solutions. Therefore, in this section, we discuss in depth the differences between FlySearch and real-life UAV control, underlining issues that were simplified or omitted in order for the problem to be addressable with existing foundational VLMs.

**Environment.**   FlySearch uses Unreal Engine 5 to provide near-photorealistic simulation of highly complex environments, with both procedurally generated and handcrafted content, using a vast set of high-resolution assets and generation rules. This setup allows for world simulation on par with the latest video games. However, the real world presents far greater complexity and unpredictability than any video game. This might lead to discrepancies between the behavior of VLM in simulated and real-world environments.

**World dynamics.**   The real world contains a vast number of dynamic elements (moving cars, walking people, flying birds, etc.) that cannot be fully simulated even with modern video game engines. Moreover, FlySearch omits some of the flight-related dynamics, such as sudden gusts of wind or abrupt weather changes, that were deemed too complex for current foundational VLMs.

**Sensor simulation.**   Our benchmark limits sensor inputs to a downward-facing camera and altitude reading with 1 m accuracy. The camera sensor is simulated using real-time ray-tracing capabilities of the engine, accounting for camera auto-exposure, depth of view, light direction (time of the day), fixed light cloud cover, and atmospheric haze. This enables an overall realistic simulation of a modern digital camera, but simplifies the weather element and omits any data loss or low-light condition emulation. Furthermore, real-life platforms may feature multiple cameras, LIDAR/RADAR, GPS units, IMU, and other sensor modalities that are not simulated in FlySearch.

**Target objects.**   The searched object is stationary (may be animated but remains in the same location), unique for the scenario (visually similar but semantically different objects may exist on the map), and visible from the top. In real-world situations, the object can move freely in the environment, including moving inside buildings, and the description of the object provided to the agent can match multiple objects. It may require more complex search strategies and the cooperation of multiple agents.

**UAV control.**   FlySearch uses high-level relative movement actions with 1 m resolution. This can be implemented in real-life UAVs using existing autopilot software such as PX4 Autopilot or DroneKit, which can accommodate wind changes and drift. However, a more direct control of the platform

could speed up the exploration process in practice and make it less dependent on GPS or inertial navigation systems. At the same time, this makes the task more complex for both VLM models and human evaluators.

**Other simplifications.** The benchmark limits agent interaction with the world to up to 20 steps due to the model context length limitations of most open-source VLMs. In real-life problems, the location of an object in a large environment may require significantly more observations. In addition, real-world UAVs are subject to hardware and software failures, none of which are simulated in FlySearch. In future work, when VLMs can solve our toughest challenge set, we plan to address some of the above limitations, making our benchmark more realistic.

## J   Example trajectories

The sample trajectories of different VLMs on FS-1 and FS-2 scenarios are provided in a separate file in the supplementary material, as well as at: `https://github.com/gmum/FlySearch/blob/main/docs/example-trajectories.pdf`

