# OpenReview forum: "FlySearch: Exploring how vision-language models explore"
_NeurIPS.cc/2025/Datasets_and_Benchmarks_Track — NeurIPS 2025 Datasets and Benchmarks Track poster_

### Official Review · Reviewer_jsiE · 2025-06-30

**Rating:** 5
**Confidence:** 3

**Summary:**

The paper introduces FlySearch, a novel benchmark designed to evaluate the exploration and navigation capabilities of Vision-Language Models in complex, photorealistic 3D outdoor environments.  Using Unreal Engine 5, the authors simulate UAV-based object search tasks across procedurally generated urban and forest scenes.  The benchmark includes three standardized scenarios (FS-1, FS-Anomaly-1, FS-2) of increasing difficulty, testing perception, contextual reasoning, and long-horizon planning.  Experiments reveal significant gaps between VLMs and human performance, particularly in systematic exploration.

**Dataset Code Accessibility:**

Yes

**Ethical Considerations:**

No, there are no or only very minor ethics concerns

**Final Justification:**

The authors resolved most of my concerns, and I have no particular issues with the method and experiments in the paper. Therefore, I raise my score.

**Limitations Weaknesses:**

The paper evaluates omitting other prominent VLM models.  Table 2 shows results for models like GPT-4o and Gemini but lacks comparisons to recent open-source VLMs optimized for navigation (e.g., RoboFlamingo, EmbodiedGPT).


While Unreal Engine 5 provides photorealism, sim2real gaps (e.g., lighting variability, sensor noise) are not discussed.  Section 3.1 describes the simulator but does not address how well it translates to real UAV deployments (e.g., wind effects, GPS inaccuracies).

The action space is overly simplified (discrete (X,Y,Z) movements), unlike real UAV control (continuous velocity, yaw adjustments).  Section 3.1 states the UAV uses an "autopilot system," abstracting low-level control.

While Appendix A mentions broader impacts, the paper does not deeply address risks (e.g., misuse of UAV navigation for surveillance).

**Strengths Contributions:**

FlySearch fills a critical gap in VLM evaluation by focusing on outdoor, embodied exploration—a setting underrepresented in existing benchmarks (e.g., Habitat, RoboTHOR are indoor-centric;  Table 1).  The use of UAVs and altitude control adds unique challenges (e.g., granularity adjustment).

The three-tiered scenario design (FS-1 to FS-2) isolates specific capabilities (perception → anomaly detection → multi-step planning), providing nuanced insights into VLM limitations (Section 4).

The paper is well-structured, with clear methodology (Section 3), detailed metrics, and prompts.

---

> ### Author Rebuttal · Authors · 2025-07-30
>
> We thank the Reviewer for their insightful comments and suggestions on how to improve the paper, as well as the praise concerning the benchmark’s usefulness and clarity. Below we address the specific comments raised by the Reviewer and provide results for additional experimentation as requested.
>
> ---
>
> ## Additional VLM evaluations
> We appreciate the Reviewer’s suggestion about evaluating embodied VLMs. In the initial version of the paper we did not evaluate embodied VLMs as in our view due to the rapid progress of the field the performance of (older) embodied VLMs is subsumed by general VLMs. To confirm our suspicion, we ran additional experiments. The 7B EmbodiedGPT in zero-shot evaluation scored 0% success rate on both the City and Forest sets of FS-1. This is consistent with other small VLMs we evaluated, such as Phi 3.5 or Llava-Interleave-7b, that quickly deteriorate in performance after the first step, and often fail to return comprehensible answers. In the case of EmbodiedGPT, in over half of the test scenarios, the experiment ended due to unparsable output after just one step.
>
> Unfortunately, we were not able to produce meaningful results for the RoboFlamingo model. During training, the model is fine-tuned to operate a robotic arm via a policy head and loses the ability to produce text output. As the action space in UAVs is vastly different, RoboFlamingo cannot be directly evaluated on FlySearch in a zero-shot setting. The only way to apply it to FlySearch would be to fine-tune the base LLM directly on FlySearch. Indeed, the representations in embodied VLMs might be more useful for fine-tuning on FlySearch than the non-embodied Qwen model that we used. While we find this avenue of research interesting, we leave this for future work.
>
>
>
> ## Sim2real gap
> We thank the Reviewer for highlighting the sim2real gap as an important concept that should be discussed in the paper. We will fix that in the revised version of the paper, briefly describing the issue in the main text and adding a detailed sim2real gap section to the appendix, providing a coherent description of the issue. We present a draft of that section below. Additionally, we would like to note that although in this paper we focus on the problem of exploring in simulation, in future work we are planning to run experiments with real-world UAVs. As such, we are very interested in the sim2real gap and seeing how the results of this paper transfer to reality.
>
> > ### Appendix: Sim2real gap
> > The main design goal of FlySearch is the evaluation of VLM exploration and 3D spatial reasoning capabilities, with UAV object navigation serving as a realistic assessment task. However, we acknowledge the fact that FlySearch can serve as a platform for testing methods designed specifically for UAV control, including non-VLM solutions. Therefore, in this section, we discuss in depth the differences between FlySearch and real-life UAV control, underlining issues that were simplified or omitted in order for the problem to be addressable with existing foundational VLMs.
> >
> > * Environment – FlySearch uses Unreal Engine 5 to provide near-photorealistic simulation of highly complex environments, with both procedurally generated and handcrafted content, using a vast set of high-resolution assets and generation rules. This setup allows for world simulation on par with the latest video games. However, the real world presents far greater complexity and unpredictability than any video game. This might lead to discrepancies between VLM behavior in simulated and real-world environments.
> > * World dynamics – The real world contains a vast number of dynamic elements (moving cars, walking people, flying birds, etc.) that cannot be fully simulated even with modern video game engines. Moreover, FlySearch omits some of the flight-related dynamics, such as sudden gusts of wind or abrupt weather changes, that were deemed too complex for current foundational VLMs.
> > * Sensor simulation – Our benchmark limits sensor inputs to a downward-facing camera and altitude reading with 1 m accuracy. The camera sensor is simulated using real-time ray-tracing capabilities of the engine, accounting for camera auto-exposure, depth of view, light direction (time of the day), fixed light cloud cover, and atmospheric haze. This enables an overall realistic simulation of a modern digital camera, but simplifies the weather element and omits any data loss or low-light condition emulation. Furthermore, real-life platforms may feature multiple cameras, LIDAR/RADAR, GPS units, IMU, and other sensor modalities that are not simulated in FlySearch.
> > * Target objects – The searched object is stationary (may be animated, but remains in the same location), unique for the scenario (visually similar but semantically different objects may exist on the map), and visible from the top. In real-world situations, the object may move freely in the environment, including moving inside buildings, and the object description provided to the agent may match multiple objects. It may necessitate more complex search strategies and the cooperation of multiple agents.
> > * UAV control – FlySearch uses high-level relative movement actions with 1 m resolution. This can be implemented in real-life UAVs using existing autopilot software such as PX4 Autopilot or DroneKit, which can accommodate for wind changes and drift. However, a more direct control of the platform could speed up the exploration process in practice and make it less dependent on GPS or inertial navigation systems. At the same time, this makes the task more complex for both VLM models and human evaluators.
> > * Other simplifications – The benchmark limits agent interaction with the world to up to 20 steps due to model context length limitations of most open-source VLMs. In real-life problems, locating an object in a large environment may require significantly more observations. Moreover, real-world UAVs are subject to hardware and software malfunctions, none of which are simulated in FlySearch.
> >
> > In future work, when VLMs can solve our hardest challenge set, we plan to address some of the above limitations, making our benchmark more realistic.
>
>
> ## Action system
> The action system is indeed simplified. This is a design decision from our side as we want to understand the general exploration and spatial reasoning capabilities of Vision-Language Models, and we omit low-level control for now. This is in line with previous research, where LLMs are often used as high-level planners and low-level actions are performed either via specialised APIs, such as in the case of Voyager [A] or a specialised controller, such as in the case DEPS [B]. The translation of high-level movement commands to low-level motor control actions is an interesting problem, although we believe that in practical scenarios, one might use existing software such as PX4 Autopilot and DroneKit, which can follow relative position control commands. We are open to exploring the additional complexity of low-level control for VLMs in future work.
>
>
> ## Broader impacts and risks
> We agree that there is a possible risk of UAV misuse. While we briefly touched on broader impacts in Appendix A, we agree that a more in-depth discussion is warranted. Our primary goal with FlySearch is to advance understanding of how VLMs explore and perceive their surroundings, independently of deployment context. However, we recognize that technologies enabling autonomous exploration, especially when integrated into UAVs, can be dual-use. In particular, VLM-driven perception systems could be misused in surveillance or privacy-invasive applications. To address this concern, we will revise Appendix A to more directly address UAV misuse and emphasize the importance of responsible research practices, below the revised text:
>
> > ### Appendix: Impact Statement
> > In this paper, we focus on the abstract problem of visual exploration – how to interact with the environment to locate the objects of interest. To evaluate this ability in VLMs, we propose a benchmark designed around flying a UAV to find objects in urban and natural environments. This is relevant to real-world applications with positive societal impact, such as search-and-rescue missions, forest fire detection, or personal assistance. However, autonomous UAVs also pose risks, as they can be misused by bad agents for surveillance or military operations. We implore users to use their best judgment for the use of the benchmark. Our benchmark is not designed for surveillance or military applications. We urge other researchers to ensure that their benchmarks (including FlySearch derivatives) and models are not directly or indirectly optimized for malicious objectives. We encourage researchers to evaluate their systems for potential biases or unintended behaviors that could lead to misuse. Finally, we would like to highlight the importance of regulatory oversight and the need for clear ethical guidelines in the deployment of autonomous UAVs.
>
> ---
>
> We hope that the answers addressed the Reviewer’s comments. We kindly ask the Reviewer to consider increasing the rating if they are satisfied with our response, and we are happy to provide further clarifications.
>
> [A] Wang, G., Xie, Y., Jiang, Y., Mandlekar, A., Xiao, C., Zhu, Y., ... & Anandkumar, A. Voyager: An Open-Ended Embodied Agent with Large Language Models. Transactions on Machine Learning Research.
> [B] Wang, Z., Cai, S., Chen, G., Liu, A., Ma, X. S., & Liang, Y. (2023). Describe, explain, plan and select: interactive planning with llms enables open-world multi-task agents. Advances in Neural Information Processing Systems, 36, 34153-34189.

---

> > ### Comment · Area_Chair_1xdg · 2025-08-05
> > **Discussion and Final Rating**
> >
> > Hi Reviewer,
> >
> > The authors have provided the rebuttal. What are your thoughts on the response? Please engage in the discussion with the authors as soon as possible, as the deadline for discussion is August 8th.
> >
> > Thanks,
> >
> > AC

---

### Official Review · Reviewer_ZW84 · 2025-07-01

**Rating:** 6
**Confidence:** 4

**Summary:**

FlySearch is a dynamic benchmark designed to assess the exploration abilities of Vision-Language Models (VLMs) in realistic, 3D outdoor environments. In its simulated environments, VLMs control a UAV and navigate to locate specified objects based on natural language descriptions.

**Dataset Code Accessibility:**

Yes

**Dataset Code Comments:**

all codes are released.

**Ethical Considerations:**

No, there are no or only very minor ethics concerns

**Final Justification:**

I have carefully checked the authors' rebuttal as well as other reviewers' opinions. After careful and comprehensive evaluation, I would like to keep my original score: strong accept.

**Limitations Weaknesses:**

1. Limited complexity in the exploration strategies: While the benchmark introduces different levels of difficulty, the exploration strategies tested in FlySearch might still be considered too simplistic for measuring more advanced, real-world exploration capabilities.

2. The paper shows that some shortcomings can be addressed with fine-tuning, but this might obscure the inherent limitations of the base models. The lack of systematic exploration capabilities might be a more fundamental flaw that can't be easily resolved by fine-tuning alone, and this needs to be further explored.  I look forward to further work after the publication of this work!

**Strengths Contributions:**

1. This work fills the gap in exploration benchmarks: This work addresses a critical gap in existing benchmarks by focusing on the exploration abilities of Vision-Language Models (VLMs). Exploration tasks in 3D outdoor environments are underrepresented in current benchmarks, and FlySearch provides a solid evaluation framework for these capabilities.

2. Comprehensive evaluation of existing models.

3. Open-source release. The creation of this benchmark involves a significant amount of work. And all the codes are released.

---

> ### Author Rebuttal · Authors · 2025-07-30
>
> We want to thank the Reviewer for their insightful and strongly positive feedback. We are delighted to hear that our work addresses a critical gap in existing VLM benchmarks, as well as that the Reviewer appreciates our commitment to open-source. Below, we address the specific comments raised by the Reviewer.
>
> ---
>
> ## Limited complexity
> We agree that real-world exploration requires more complex behaviors than the ones examined in this paper. However, the benchmark is tailored to the capabilities of the current VLMs, which perform reasonably well on FS-1 but largely fail on FS-2. In the future, as VLMs improve, we imagine adding new levels of complexity, e.g., by significantly extending the search area, adding additional degrees of freedom (UAV and camera rotation), or adding more difficult weather settings. Moreover, our public codebase allows other researchers to adapt FlySearch to their requirements.
>
> ## Limitations of fine-tuning
> We agree with the reviewer that fine-tuning will not work if strong exploration and spatial reasoning capabilities are not present already in the underlying models. We are looking forward to examining this problem further in future work.
>
> ---
>
> We hope that our response addresses the Reviewer’s comments, and we are happy to provide further clarification.

---

### Official Review · Reviewer_zXoH · 2025-07-02

**Rating:** 5
**Confidence:** 3

**Summary:**

This paper introduces FlySearch, a photorealistic 3D outdoor benchmark designed to evaluate VLMs in active, goal-driven exploration tasks. Unlike prior work focused on static or curated settings, FlySearch simulates messy and unstructured real-world environments, requiring agents to search for and navigate to target objects. The benchmark includes three levels of scenario difficulty, and experimental results show that even state-of-the-art VLMs struggle with the simplest tasks, performing far below humans—especially as task complexity increases.

**Dataset Code Accessibility:**

Yes

**Ethical Considerations:**

No, there are no or only very minor ethics concerns

**Final Justification:**

The author rebuttal has addressed my concerns.

**Limitations Weaknesses:**

1. The dataset includes only two main environments—forest and city. Introducing more varied environments to better assess the robustness of VLMs may be beneficial.

2. Each environment has only a few categories. Expanding the object vocabulary may better simulate real-world challenges.

**Strengths Contributions:**

1.The paper is well-written and easy to follow.

2.The construction process of the FlySearch benchmark is clearly stated.

3.Compared with other datasets, this benchmark satisfies a broader set of characteristics, enabling effective evaluation of VLMs in outdoor navigation scenarios.

---

> ### Author Rebuttal · Authors · 2025-07-30
>
> We want to thank the Reviewer for their insightful and positive feedback. We are happy to hear that the Reviewer found our benchmark as enabling effective evaluation of VLMs, as well as the paper to be well-written. Below, we address the specific comments raised by the Reviewer
>
> ---
>
> ## Number of environments
> We agree that introducing more varied environments would allow us to better assess the robustness of VLMs. However, preparation of a high-quality, large, open-world environment with procedurally generated elements and semantically relevant objectives requires a significant amount of time and resources. Therefore, for the current set of challenges, we’ve decided to focus on urban (city) and natural (forest) settings, which allow us to analyze VLM exploration capabilities in vastly different scenarios. That said, in future work, we consider incorporating a broader range of environments (e.g., industrial, desert, or underwater). Moreover, our public codebase allows anyone with experience in Unreal Engine to add their own maps to the benchmark. We are working on creating more detailed documentation, including a tutorial on adding environments. We appreciate the Reviewer’s suggestion, and we hope that we can extend the benchmark in the future, either ourselves or with the help of the community.
>
> ## Number of categories
> We agree that increasing the number of categories would further increase the usability of the benchmark. However, for the time being, we decided to start with a smaller set of well-curated objects. When selecting object types for each environment we focused on four criteria: a) they had to be easy to recognise visually by humans and VLMs, b) their name had to be unambiguous (not matching other objects in background), c) they had to semantically fit the environment (or not fit in case of the anomaly benchmark), and d) ideally there should be some common-sense knowledge about where they can be found (e.g. a car is likely located on a road, or a forest fire can be found by following smoke). For those reasons, we decided to have a shorter, highly curated list of objects. However, just as in the case of adding maps, our public codebase allows anyone to add additional objects. We will also consider adding more objects in future upgrades of our benchmark.
>
> ---
>
> We hope that our response addresses the Reviewer’s comments, and we are happy to provide further clarification.

---

> > ### Comment · Reviewer_zXoH · 2025-08-05
> >
> > Thank you for the clarification. The authors' rebuttal has addressed my concerns.

---

### Official Review · Reviewer_iojp · 2025-07-14

**Rating:** 4
**Confidence:** 3

**Summary:**

This paper provides a benchmark to evaluate how VLMs search and navigate to objects in complex scenes. Specifically, the authors leverage Unreal Engine 5 as the simulator to generate scenarios of searching for objects in city and forest environments. A controller is used to communicate with VLMs and the simulator and return the results. Three standardized challenges, FS-1, FS-Anomaly-1, and FS-2 are provided for easy scenarios, anomaly detection, and harder scenarios (potentially obscured objects). Experiments show that VLMs underperform compared to human baseline.

**Dataset Code Accessibility:**

Yes

**Ethical Considerations:**

No, there are no or only very minor ethics concerns

**Final Justification:**

While I feel that the current version still requires some refinement, I do not object to the paper being accepted. I will maintain my decision as borderline accept.

**Limitations Weaknesses:**

- The distinction between FlySearch and other UAV-centric outdoor VLN environments, such as TRAVEL, CityNav, or AerialVLN, is unclear. Additionally, only one reference ([61]) considers this software's application. Why is the application of FlySearch necessary, and how does it differ from [61]?

- Why not fine-tune on various scenarios? If fine-tuning fails to address the issues in even the simplest exploration tasks, it suggests that VLMs might be inadequate for this task. Consequently, this benchmark could be deemed unnecessary.

- Please clarify the evaluation metrics used in this benchmark. A detailed explanation would help in assessing the validity of the experiment results. The term "while the object is visible" lacks clarity. Please provide a precise definition or criteria for this condition.
Why was altitude difference chosen over Euclidean distance for measurement?

- The arrangement of this paper needs improvement. The placement of tables and figures is not optimal, which can disrupt the flow of reading.

**Strengths Contributions:**

- The benchmark settings are interesting and technically sound, and the dynamic benchmark is novel to me.
- The paper is easy to follow, and the tables and figures are clear and well-illustrated.
- The benchmark includes various baselines, human studies, different scenarios and tasks.

---

> ### Author Rebuttal · Authors · 2025-07-30
>
> We want to thank the Reviewer for their insightful and positive feedback. We are happy to hear that the Reviewer found our benchmark to be interesting and novel, the paper to be easy to follow, and the work to be technically sound. Below, we address the specific comments and provide results from the additional experiments we ran at the request of the Reviewer.
>
> ---
>
> ## FlySearch vs other benchmarks
> The main distinction between FlySearch and TRAVEL, CityNav, and AerialVLN lies in the core task (Object Navigation vs. Vision-Language Navigation) and, therefore, the evaluated model capabilities. The papers referenced here focus on Vision-Language Navigation, i.e., instruction following for moving from point A to point B. On the contrary, we focus on ObjectNav, which requires the agent to find the object in question without detailed instructions. As noted by [66]:  “A typical VLN agent receives a (sequence of) language instruction(s) from human instructors at a designated position [...]”
>
> AerialVLN [32] is an example of a VLN task with high-level UAV control and a simulated environment, which follows the above definition. TRAVEL [57] aims to make drone-based VLN more low-level by modifying the agent’s action space for direct UAV control. CityNav [26] introduces a VLN environment based on scans of real-life cities from SensatUrban [A] (instead of simulating them). Papers referenced above focus on instruction following in navigation (i.e., VLN) while our benchmark puts emphasis on finding objects without further instructions (i.e., ObjectNav).
>
> ObjectNav is dominated by indoor environments, with the OUTDOOR [61] mentioned by the Reviewer being a notable exception, as it considers using the Microsoft AirSim plugin [50] to simulate a drone. However, even though the paper considers using UAVs, the action space is still 2-dimensional (horizontal movement only), while FlySearch has a 3-dimensional exploration space (the UAV can also move up and down).  This emphasises dynamic altitude control to manage uncertainty and avoid redundant exploration, as the level of visual detail and the size of the visible area varies depending on the altitude. As such, success in FlySearch requires an efficient, emergent search strategy, where the agent must reason about where the object is likely to be, when looking from high altitude. Overall, this is a much more complex problem, requiring pre-existing knowledge of the real world to understand the contextual cues to exploit a wide field of view at high altitudes and fly low only in promising areas. We will update the related work section to better highlight the difference between FlySearch, VLN benchmarks, and OUTDOOR.
>
> ## Impact of fine-tuning
> We agree that the behavior of VLMs after fine-tuning is important to check. As such, we ran additional experiments during the rebuttal, fine-tuning Qwen2.5-VL 7b on City on a different set of scenarios than the test set. The model got a 51.4% success rate, a vast improvement compared to Qwen’s 1.5% zero-shot performance and 41.5% performance of the Gemini 2.0 Flash (best zero-shot model on the City environment). Additionally, in Table 2 of the current paper revision, we provide the result of fine-tuning a model directly on the Forest task. Here, we also observed a substantial improvement – 57% for fine-tuned Qwen2.5-VL 7B (vs 52% of the best proprietary model in zero-shot evaluation). We believe that if we were to fine-tune a bigger model, the performance would rise further, although such experiments are currently beyond our computational capabilities. Finally, we agree that today's VLMs may not be able to solve the exploration problems discussed in this paper. However, given the rapid pace of advancement, we believe this will change in the future. We hope our benchmarks will be a useful tool for measuring the progress of VLMs in solving these types of problems.
>
>
> ## FlySearch metrics
> The metrics we use are inspired by the ObjectNav literature, although we introduce some changes based on our empirical investigation of the model’s performance. To check whether the agent can see the object, we calculate the agent's cone of view and check whether the object's center is in the agent’s view cone, as described in the Appendix (Section E – Success criterion implementation). We use the height requirement, since the agent knows their distance from the ground (it is provided in context after every move) while the Euclidean distance to the target might be more difficult to estimate for both VLMs and human evaluators. This leads to a simple decision rule – if I see the object, and I’m at an altitude equal to or lower than 10 meters, I can say that the object was found. We thank the Reviewer for this suggestion, and we will add a more thorough discussion of the metrics in the main text.
>
> ## Paper arrangement
> We agree that the arrangement of the paper can be improved, and we will fix it in the camera-ready version if accepted, as then we will have one more page to fit in our results. Then, we will reposition the figures and tables to improve readability.
>
> ---
>
> We hope that the answers addressed the Reviewer’s comments. We kindly ask the Reviewer to consider increasing the rating if they are satisfied with our response, and we are happy to provide further clarifications.
>
> [A] Hu, Q., Yang, B., Khalid, S., Xiao, W., Trigoni, N., & Markham, A. (2022). Sensaturban: Learning semantics from urban-scale photogrammetric point clouds. International Journal of Computer Vision, 130(2), 316-343.

---

> > ### Comment · Area_Chair_1xdg · 2025-08-05
> > **Discussion and Final Rating**
> >
> > Hi Reviewer,
> >
> > The authors have provided the rebuttal. What are your thoughts on the response? Please engage in the discussion with the authors as soon as possible, as the deadline for discussion is August 8th.
> >
> > Thanks,
> >
> > AC

---

### Decision · Program_Chairs · 2025-09-18

**Decision:**

Accept (poster)

**Comment:**

This paper was reviewed by four experts in the field. The recommendations are (Strong Accept, Accept x 2, Borderline Accept). Based on the reviewers' feedback, the decision is to recommend the acceptance of the paper. The reviewers did raise some valuable concerns (especially more statements about the distinction and comparison between FlySearch and other works raised by Reviewers iojp, zXoH, and jsiE, manuscript further polishing and refinement raised by All Reviewers) that should be addressed in the final camera-ready version of the paper. The authors are encouraged to make the necessary changes to the best of their ability.